# Virtual reality as a strategy for intra-operatory anxiolysis and pharmacological sparing in patients undergoing breast surgeries: The V-RAPS randomized controlled trial protocol

Joe Zako [1,2*], Nicolas Daccache[1,2], Julien Burey[3,4], Ariane Clairoux[3], Louis Morisson[1,3], Pascal Laferrière-Langlois [1,3]

1 Maisonneuve-Rosemont Hospital Research Center, Montreal, Quebec, Canada, 2 Faculté de médecine, Université de Montréal, Montreal, Quebec, Canada, 3 Department of Anesthesiology and Pain Medicine, Maisonneuve-Rosemont Hospital, Centre Intégré Universitaire de Santé et Services Sociaux de l'Est de L'Île de Montréal, Montreal, Quebec, Canada, 4 Department of Anesthesiology and Critical Care Medicine, Tenon Hospital, APHP Sorbonne Université, Paris, France

* joe.zako@umontreal.ca

## Abstract

### Introduction

Virtual reality (VR) has increasingly found applications beyond leisure and video games, extending into the field of medicine. Recent studies indicate that VR can effectively reduce anxiety and pain in pediatric patients undergoing uncomfortable medical procedures, such as burn wound care. Yet, VR use in the operating room is still rare, despite a growing trend toward regional anesthesia without general anesthesia; physicians still frequently rely on pharmacological sedation to manage procedural anxiety. By leveraging VR's anxiolytic properties, it may be possible to decrease the need for intravenous (IV) sedation which is associated with risk of adverse events like apnea and hypoxemia and delayed recovery.

### Objectives

This study's main objective is to determine the impact of VR on IV sedation requirements in adult patients undergoing breast surgery under paravertebral (PV) block without general anesthesia. We will be using Paperplane Therapeutics' VR headset and software which offer three distinct VR scenario choices. We hypothesize that VR immersion will reduce the need for intraoperative pharmacological sedation. Secondary objectives include assessing the tolerance of patients to the VR headset, examining the impact of the chosen VR scenario on the primary outcome, evaluating the incidence of adverse effects, measuring patient satisfaction, and analyzing the output of the Nociception Level (NOL) Index among awake surgical patients.

**Data availability statement:** No datasets were generated or analysed during the current study. All relevant data from this study will be made available upon study completion.

**Funding:** The author(s) received no specific funding for this work.

**Competing interests:** Dr PLL declares ownership interest in private companies unrelated to this work (Divocco Medical and Divocco AI). Other authors declare no competing interests.

## Materials and methods

This single center randomized controlled trial will enroll 100 patients aged 18 or above undergoing breast surgery under PV block. Participants will be randomly allocated to the VR group or the control group; both will have access to pharmacological sedation through patient-controlled sedation (PCS). Participants in the VR group will choose between three different VR scenarios and will be allowed to switch between these scenarios during surgery. The primary outcome will be the time-adjusted and weight adjusted dose of self-administered intraoperative propofol. Secondary outcomes will include patient satisfaction, adverse events, and post-anesthesia care unit length of stay (PACU LOS).

## Ethics

This trial has been approved by the regional ethics committee (Comité d'Éthique de la Recherche du CIUSSS de l'Est de l'Île de Montréal) on September 9th, 2024.

## Trial registration

ClinicalTrials.gov (July 25th, 2024). Unique protocol ID: 2025–3802. Trial identification number: NCT06522711.

---

## 1. Introduction

### 1.1. Background and rationale

Virtual reality (VR) is a novel technology that operates through computer generated three-dimensional environments with which the user can interact. These virtual experiences are accessible on various interfaces, ranging from smartphone screens that display a VR environment to fully immersive headsets and systems that provide haptic feedback, simulating the sense of touch [1]. Over the past decade, the use of VR has expanded across various medical fields. This technology has been used to train residents to perform medical procedures, to allow experienced surgeons to rehearse complex surgeries and has even been used as a substitute for exposure therapy when treating patients with specific phobias or anxiety disorders [2–5].

In anesthesiology, VR immersion has been utilized during various procedures, including orthopedic surgeries of the upper extremities, hip and knee arthroplasties under regional anesthesia, dental care, burn wound treatment, and even during anesthesia induction [6,7]. It has shown significant potential in reducing anxiety and improving patient comfort in these perioperative and periprocedural contexts, although the literature on the topic mostly pertains to pediatric populations [8,9]. Notably, research indicates that interactive, gamified VR scenarios provide significantly greater anxiolysis compared to passive scenarios, likely due to their increased distractibility [10]. While a reduction of procedural anxiety has also been observed in adult populations undergoing VR immersion [11], the assessment of objective outcomes, such as procedural sedative usage, has not been thoroughly explored.

Pharmacological sedation is a crucial part of awake medical procedures, primarily aimed at providing anxiolysis and, when necessary, amnesia. However, these medications can have serious side effects, such as respiratory depression, especially when used alongside opioid analgesics [12]. In fact, research suggests a dose-dependent relationship between the amount of procedural sedatives administered and the length of recovery times in the post-anesthesia care unit (PACU) [13]. This highlights a critical gap in the current literature, as VR could be a low-risk adjunct or alternative to procedural sedation. This is especially relevant for breast surgeries performed only under regional anesthesia, which is increasingly recognized as a valid alternative to general anesthesia. Notably, this approach has been shown to be equivalent to general anesthesia in terms of breast cancer recurrence and persistent incisional breast pain, but often necessitates substantial sedative use, creating an opportunity for VR to potentially reduce sedation requirements [14].

Finally, the Nociception Level (NOL) index is a technology developed by Medasense Biometrics Ltd (Ramat Gan, Israel) [15], which utilizes artificial intelligence algorithms to provide nociception scores ranging from 0 to 100. Using a non-invasive finger probe connected to the PMD-200 monitor, it evaluates photoplethysmography, galvanic skin response, accelerometry, and skin temperature to generate its output. While the NOL index has been validated for use in anesthetized patients undergoing surgery [16], its application in awake surgical populations remains largely unexplored. Investigating its use in this setting is relevant for several reasons. Firstly, awake patients under regional anesthesia with sedation may still experience nociception or discomfort, even if they do not fully verbalize it. Secondly, correlation between NOL trends and patient distress or sedation self-administration could suggest a role for NOL as an early indicator for sedation adjustments. Finally, studying the NOL index in awake patients offers a unique opportunity to compare objective nociception data with real-time subjective patient feedback.

### 1.2. Objectives and hypotheses

The primary objective of this study is to determine if intraoperative virtual reality immersion reduces the use of self-administered propofol for patients undergoing breast surgery under paravertebral (PV) block.

Our secondary objectives are the following

- Determine the participants' initial enthusiasm at the idea of using a VR headset during surgery.

- Evaluate the level of anxiety and information requirement before the surgery, as they could act as potential confounders if randomization does not adequately balance groups;

- Evaluate the incidence of adverse effects such as cybersickness, nausea, bradycardia, desaturation, and hypotension;

- Evaluate the time the patient spent wearing the headset;

- Evaluate the differences in sedation requirements depending on the type of VR scenario.

- Evaluate need for additional local anesthesia;

- Evaluate quantities of fentanyl administered;

- Evaluate quantities of ketamine administered;

- Evaluate the requirement of post-operative care and the post-anesthesia monitoring time;

- Determine the ease of use of the technology, enjoyment of the first scenario chosen and overall satisfaction with the experience.

This trial also has an exploratory objective, which is to investigate the use of the Nociception Level (NOL) index in awake patients undergoing surgery, and the ability to anticipate sedation self-administration via the NOL index.

Regarding our primary objective, we hypothesize that patients undergoing breast surgery under PV block with an immersive VR experience will self-administer less propofol than the control group. We also hypothesize that interactive scenarios will further reduce the requirement for sedation, when compared to non-interactive scenarios.

### 1.3. Trial design

This is the protocol for an open-label, single-center randomized controlled trial.

## 2. Methods

This protocol follows the Standard Protocol Items: Recommendations for Interventional Trials (SPIRIT) Statement guidelines [17].

### 2.1. Participants, interventions, and outcomes

**2.1.1. Study setting.** This trial will be conducted at Maisonneuve-Rosemont Hospital, a part of the Centre intégré universitaire de santé et de services sociaux (CIUSSS) de l'est de l'île-de-Montréal (CEMTL), located in Montreal, Quebec, Canada.

**2.1.2. Eligibility criteria.** We will recruit consenting adult patients (≥ 18 years) undergoing elective breast surgery under PV block without general anesthesia.

Patients will automatically be excluded from the study if they have any of the following:

1. Hearing or visual impairment;

2. History of epilepsy, seizure, or severe dizziness;

3. Cognitive or psychiatric impairment that would interfere with informed consent, patient-controlled sedation (PCS) use, or VR engagement, as determined by:

   a. A documented clinical diagnosis in the patient's medical record, or

   b. A Montreal Cognitive Assessment (MoCA) test, administered if significant clinical doubt exists;

4. Recent eye or facial surgery or wounds;

5. Inability to use the VR hand controller;

6. Allergy to one of the protocolized drugs.

**2.1.3. Interventions.** Prior to entering the operating room, all participants will receive an explanation on the use of intraoperative PCS. For those in the intervention group, a short video will introduce all three VR scenarios, after which participants will select their preferred scenario, which will then be documented.

Upon arrival in the operating room, participants will undergo standard monitoring, including non-invasive blood pressure measurement, pulse-oximetry, and continuous electrocardiography using the Dräger Infinity C700 monitor (Dräger Medical, Lübeck, Germany). Throughout the entire duration of surgery, the NOL index finger probe, which is connected to the PMD-200 monitor (Medasense Biometrics Ltd, Ramat Gan, Israel) will be applied. All intraoperative data and events will be recorded on the research computer.

The attending anesthesiologist will perform PV block before surgery according to their usual practice. Only anesthesiologists with a minimum of 10 PV blocks performed within the last year will be qualified to administer this procedure to trial participants. Standardized doses of intravenous (IV) sedation (0.15 mg/kg of propofol) and IV analgesia (0.5 mcg/kg of fentanyl) will be administered to all participants prior to realization of the PV block and will be repeated if necessary. Once the block is complete, the participant is positioned supine, and the surgeon is ready for disinfection, the VR headset

will be applied to those in the intervention group and their chosen scenario will begin. A dedicated research staff member will be present in the operating room throughout the procedure to manage all aspects of the VR intervention, including assisting with headset application, adjusting the VR scenarios upon patient requests, and addressing any technical issues or patient discomfort related to the VR system. Notably, the interactive VR scenarios do not require significant hand or arm movement, as they rely solely on eye-tracking software to aim within the game. The only manual action required is pressing the controller trigger at specific time points. If at any time the patient expresses a desire to discontinue VR during the surgery, we will first offer them an alternate scenario. If they reiterate their willingness to stop, with or without having experienced the new scenario, the headset will be removed for the remainder of the surgery. In case of adverse events attributable to VR gear, such as cybersickness, discontinuation of VR immersion will be at the discretion of the anesthesiologist as well as the patient.

The VR headset to be used is the commercially available Paperplane Therapeutics VR system, specifically designed for medical applications. It offers a fully immersive experience incorporating both visual and auditory elements; notably, it is equipped with integrated near-ear speakers positioned along the side branches of the headset. These speakers are designed to direct sound toward the patient's ears, providing clear audio at a volume that is perceptible to the patient but remains low enough to avoid disturbing the surgical team or others in the operating room. This design also ensures that patients can continue to communicate freely with clinicians throughout the procedure. A visual representation of the VR device used as well as the possible scenario choices can be seen in Fig 1.

In both groups, the PCS protocol will consist of 0.15 mg/kg boluses of IV propofol with a lockout period of 2 minutes; additional clinician boluses may be administered based on the lead anesthesiologist's judgment. For breakthrough pain, local anesthesia can be utilized by the surgical team. The anesthesiologist can also administer 0.5 mcg/kg doses of IV fentanyl and,

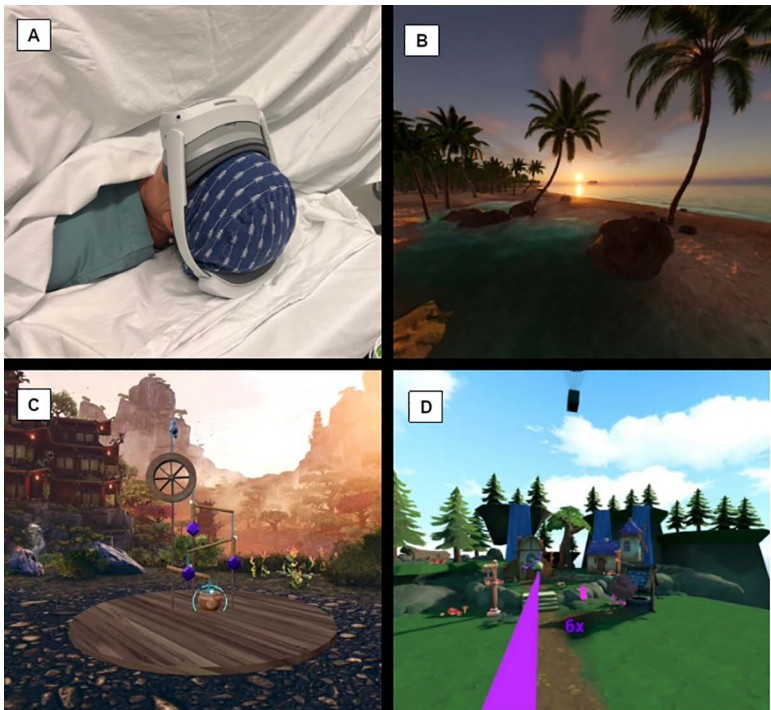

**Fig 1. VR headset and scenario choices.** A Paperplane Therapeutics VR headset; B Scenario 1 – Calm; C Scenario 2 – Puzzle; D Scenario 3 – Action.

if insufficient, 0.15 mg/kg doses of IV ketamine. If one of the following criteria are met, the anesthesiologist will switch the participant to general anesthesia: inadequate surgical anesthesia despite attempting all previous measures, persistent hemodynamic or respiratory instability (mean arterial pressure <60 mmHg despite fluid resuscitation and vasopressors or $SpO_2$ <92% that does not respond to non-invasive oxygenation methods) and any surgical complication requiring general anesthesia.

**2.1.4. Outcomes.** The primary outcome is the time-adjusted and weight-adjusted average or median self-administered dose of propofol in mcg/kg/min.

Secondary outcomes include

- Initial enthusiasm at the idea of using a VR headset during surgery assessed by research staff on a 10-point Likert scale for all participants, prior to intervention allocation as part of a questionnaire;

- Level of anxiety and information requirement measured preoperatively by research staff using the Amsterdam Preoperative Anxiety and Information Scale (APAIS) [18];

- Incidence of intraoperative and postoperative adverse events such as bradycardia (heart rate <50 bpm), desaturation ($SpO_2$ <92%) and hypotension (MAP <60) as well as subjective adverse events, such as cybersickness or nausea documented intraoperatively by the anesthesiologist and research staff based on real-time clinical observation or patient self-reporting;

- Average or median total time, in minutes, during which the VR headset was worn, as well as the percentage of that time relative to the overall duration of the surgery, recorded by dedicated research staff in the operating room using a time log;

- Average or median total duration in minutes spent by the patient on the VR scenario chosen, and the order in which VR scenarios were presented, recorded by dedicated research staff in the operating room using a time log and a case report form (CRF);

- Percentage of patients that switched scenarios;

- Percentage of patients that removed the headset before the end of the surgery;

- Percentage of patients requiring additional local anesthesia;

- Weight-adjusted average or median administration of intraoperative IV fentanyl in mcg/kg;

- Weight-adjusted average or median administration of intraoperative IV ketamine in mg/kg;

- PACU length of stay (LOS) measured from PACU admission to PACU discharge, based on the PACU nursing records;

- Ease of use of the technology and enjoyment of the first scenario chosen assessed post-operatively on a 10-point Likert scale in the intervention group only as part of a postoperative questionnaire administered by dedicated research staff;

- Overall satisfaction with the experience assessed post-operatively on a 10-point Likert scale in both groups as part of a postoperative questionnaire administered by dedicated research staff.

Exploratory outcomes

- NOL index readings over time per patient, recorded continuously intraoperatively via a finger probe, automatically stored and extractable by dedicated research staff from the PMD-200 monitor with precise timepoints. NOL index spikes will be defined;

- PCS bolus administration timepoints, automatically recorded by the pump device and extracted by dedicated research staff.

Exploratory analysis will focus on evaluating NOL trends over time and assessing whether NOL peaks are associated with PCS administration timepoints.

**2.1.5. Participant timeline.** The research team will review the elective surgical schedule at Maisonneuve-Rosemont Hospital to identify eligible patients at least one week before their surgery. After institutional consent is obtained, patients' medical charts will be screened based on the study's eligibility criteria. Potential participants will be contacted by telephone to explain the project and answer supplementary questions.

One their interest in the project is confirmed, candidates will meet with the research team on the day of their surgery to address any further queries, sign the consent form, and complete the standard preoperative questionnaire. Participants will undergo randomization following regional anesthesia and will receive surgery, with or without the VR headset, based on their allocation. A variety of preoperative, intraoperative and postoperative assessments relating to our outcomes of interest will be performed. Their involvement in the study will conclude upon discharge from the PACU. A detailed schedule of enrollment, interventions, and assessments is provided in Fig 2.

**2.1.6. Sample size.** The available literature on sedation requirements for adult patients using intraoperative VR headsets during surgeries under regional anesthesia is limited and heterogeneous, with studies reporting up to 93% reductions in sedation requirements [19], and others reporting no significant changes [20]. However, in a study evaluating lidocaine as an adjunct to reduce propofol requirements during pediatric colonoscopy, the lidocaine group consumed 35.5% less propofol than the control group, which was associated with significantly shorter awakening times, faster recovery times, and fewer involuntary movements during the procedure [21]. Similarly, a study comparing PCS to clinician-controlled sedation found a 39.3% reduction in propofol usage in the PCS group, which correlated with fewer episodes of deep sedation, fewer airway and breathing adverse events, and faster recovery times [22]. Therefore, for our sample size calculation, we estimated a 30% reduction of propofol usage in the VR group compared to the control group to be an approximate threshold for clinical significance. A two-sample t-test was used for the calculation, with a two-sided hypothesis test and a significance level of 0.05. To achieve 80% power, 90 participants were required, with the sample size inflated to 100 to account for attrition (e.g., dropout, protocol deviations).

Paravertebral block with propofol sedation has been validated as a safe and effective alternative to general anesthesia for breast cancer surgery, offering benefits such as higher turnover rates, reduced postoperative nausea, and faster hospital discharge [23]. According to historical data by Sessler et al., patients undergoing breast surgery with paravertebral block received a median propofol dose of 525 mg per patient [14].

**2.1.7. Recruitment.** Recruitment for this trial is anticipated to begin on October 30th, 2024. By screening all potential candidates and contacting them by telephone beforehand, we will ensure optimal participant enrolment. We expect to complete recruitment by September 2026. Experimentation and data collection will also occur throughout this period.

## 2.2. Assignment of interventions

Electronic randomization of the participants will be performed by an independent third party, using the National Cancer Institute's Clinical Trial Randomization tool [24]. Specifically, we will randomize participants in a 1:1 ratio using a maximal randomization method, allowing a maximum tolerated imbalance of 3. The randomization sequence will then be implemented in the REDCap application (Vanderbilt University) [25]. Each participant's allocation will be sealed in an opaque envelope and handed to the dedicated research staff not involved in patient care. The envelope will be opened at the patient's entry into the operating ward, after confirming that the surgery will be performed under regional anesthesia. The intervention will not be blinded to participants or personnel after allocation.

## 2.3. Data collection, management and analysis

**2.3.1. Data collection methods.** We will collect data at baseline, intraoperatively and postoperatively to assess primary and secondary outcomes. Participant socio-demographic information, medical history and current medication,

| | **Pre-Allocation** | **Allocation** | **Post-allocation** | **Close-out** |
|---|:---:|:---:|:---:|:---:|
| **TIMEPOINT** | Before allocation | Preoperative | Intraoperative | PACU discharge |
| **ENROLMENT** | | | | |
| Eligibility screen | ▨ | | | |
| Telephone outreach | ▨ | | | |
| Informed consent | ▨ | | | |
| Allocation through randomization | | ▨ | | |
| **INTERVENTIONS** | | | | |
| VR immersion | | | ▨ | |
| Control | | | ▨ | |
| **OUTCOME ASSESSMENTS** | | | | |
| Initial enthusiasm at the idea of using VR | ▨ | | | |
| Pre-operative anxiety | | ▨ | | |
| Adverse events | | | ▨ | |
| Scenario switches | | | ▨ | |
| VR usage time | | | ▨ | |
| VR gear removal | | | ▨ | |
| Propofol usage | | | ▨ | |
| Local anesthetic usage | | | ▨ | |
| Fentanyl usage | | | ▨ | |
| Ketamine usage | | | ▨ | |
| NOL index | | | ▨ | |
| PACU LOS | | | | ▨ |
| Patient satisfaction | | | | ▨ |

**Fig 2. Schedule of enrolment, interventions and assessments.**

will be collected from their medical chart. Their education level and familiarity with VR technology will also be recorded through a preoperative questionnaire. Pre-operative anxiety levels will be assessed using the APAIS, a six-item questionnaire designed to measure both anxiety and the desire for information before surgery. The scale provides an overall anxiety score ranging from 4 to 20 and an information-seeking score ranging from 2 to 10, with higher scores indicating greater preoperative anxiety and a stronger desire for information, respectively. A postoperative questionnaire will be used to collect data on patient satisfaction. It is of note that, aside from the APAIS, both our preoperative and postoperative questionnaires are self-composed and have not yet undergone formal scientific validation.

NOL index data from the PMD-200 medical monitor will be automatically collected by the device. Other intraoperative events, such as propofol self-administration, as well as the use of fentanyl or ketamine, will be manually recorded. Post-operative outcomes will be collected through questionnaires administered to participants prior to their discharge from the PACU, at which point we will also record PACU length of stay (LOS). Adverse events of any kind will be documented when spontaneously reported or observed clinically.

We will monitor and document the number of participants who withdraw from the study. If a participant wishes to discontinue their involvement, they will be removed from the study at their request, and no further data will be collected or retained after their withdrawal.

**2.3.2. Data management.** Data from this study will be collected and managed using REDCap electronic data capture tools hosted at Maisonneuve-Rosemont hospital. Research staff will systematically double-check all data entries for completeness and accuracy before final submission. Each participant will be assigned a unique identification number upon enrollment to maintain confidentiality and ensure anonymization.

All collected data will be securely stored on a designated research computer, which will remain offline and be located in a restricted area in Maisonneuve-Rosemont hospital. Notably, this computer will utilize the integrated event-tagging system of the PMD-200 or the BetterCare software (provided by Dräger, Lübeck, Germany), which is connected to the Dräger anesthesia workstation. Usage of this computer will be limited to authorized research personnel, who will need encrypted logins to access study data.

To promote data quality, double data entry will be implemented for our primary outcome. Furthermore, regular, informal audits will be conducted by the research team to verify the accuracy of the study data.

After study completion, electronic data will be retained on secure, encrypted servers for a period of 7 years, as per local institutional and ethical guidelines. Physical copies of patient information, such as consent forms, will be securely stored in a locked file cabinet in the anesthesiology department.

**2.3.3. Statistical methods.** Descriptive statistics will be used to summarize the data by group. Primary outcome and secondary continuous parameters will be presented as means with standard deviations, or medians with interquartile ranges if the data is skewed or non-normally distributed. Categorical variables will be reported as frequencies (%). 95% confidence intervals (CIs) for proportions or mean/median differences will be presented based on the type of endpoint analyzed, and statistical significance will be determined using an alpha level of 0.05. Parametric, two-sample t-tests will be performed for normally distributed outcomes. Otherwise, a non-parametric Wilcoxon test will be performed.

Univariate analyses will be performed to explore relationships between the primary outcome (total propofol consumption) and other potential variables. Variables showing significant associations ($P < 0.05$) or clinical relevance will be considered for inclusion in a multivariable linear regression model. To evaluate the fit of our linear regression models, we will examine $R^2$ and adjusted $R^2$ values to assess explanatory power. In addition, residual plots and diagnostic statistics will be used to evaluate assumptions of linearity, normality, and homoscedasticity. If model fit is inadequate, appropriate transformations or alternative modeling strategies will be considered. Logistic regression may be employed where appropriate for binary secondary or exploratory outcomes. To evaluate the fit of our logistic regression models, we will conduct the Hosmer-Lemeshow goodness-of-fit test; $P > 0.05$ will be considered indicative of an adequate model fit. If poor fit is detected, alternative modeling strategies, such as transformations or interaction terms, will be considered.

Subgroup analysis will also be performed to explore how the selected initial VR scenario may impact our primary outcome. In the case of conversion to general anesthesia, the participant will be considered a protocol deviation. These participants' data will still be included in the final analysis following an intention-to-treat (ITT) approach. Additionally, we will compare the prevalence of protocol deviations between the intervention and control groups to assess whether the conversion rate differs between conditions. To minimize protocol deviations, patient recruitment and randomization will only be finalized after the paravertebral block has been performed and its effectiveness has been confirmed by the lead clinician. If the block is deemed grossly ineffective, the patient will not be recruited or randomized.

As part of our exploratory analysis, we will investigate the relationship between the NOL index and propofol self-administration to determine whether NOL spikes serve as a predictor of subsequent propofol boluses. To achieve this, we will define a spike as any NOL score exceeding a threshold of 50 for a pre-specified duration of at least 10 seconds. We will then use time-series analysis to assess the temporal relationship between NOL spikes and propofol bolus administration. This will allow us to apply logistic regression and/or time-to-event analysis to determine whether NOL spikes significantly predict the likelihood of a propofol bolus within a predefined timeframe of up to 60 seconds.

All statistical analyses will be conducted using SAS, SPSS, or Python programming language with VS Code or Jupyter Notebook software.

## 2.4. Monitoring

**2.4.1. Data monitoring.** The study is considered low-risk and will not require a formal data monitoring committee. The principal investigator (PI) or designated personnel will regularly review the data on a monthly or bi-weekly basis to assess study completeness, enrollment progress, protocol deviations, participant dropouts, and adverse events. The study will also be continuously overseen by the regional ethics committee.

**2.4.2. Potential harms.** The devices used in the study include a VR headset wirelessly paired with a tablet for broadcasting immersive scenarios. These devices do not interfere with intraoperative monitoring, and the risk of adverse events associated with their use is very low.

Any incidence of adverse effects, spontaneously reported or directly observed, will be documented and assessed as part of our secondary outcomes.

**2.4.3. Auditing.** There is no formal plan for auditing or inspection in this study.

## 3. Ethics and dissemination

### 3.1. Research ethics approval

This protocol has been approved by the regional ethics committee (Comité d'éthique en recherche du CIUSSS de l'Est de l'Île de Montréal) on September 9th, 2024.

### 3.2. Protocol amendments

Any changes to the study protocol will be submitted for review by the regional ethics committee and updated on ClinicalTrials.gov. These amendments will be communicated to participants and study personnel where necessary and will be documented in the final study report. In the event of study termination, participants will be notified, and data will be handled according to ethical guidelines.

### 3.3. Consent or assent

During recruitment, potential participants will be contacted by the research team via phone to explain the project. The communication script used during these calls has been approved by the regional ethics committee. Prior to surgery, the research team will meet with candidates in person to answer any remaining questions. If participants wish to proceed, informed consent will be obtained and signed. The form used to obtain written consent by the research staff is available in the appendices (S1 Appendix).

### 3.4. Confidentiality

Protected health information will not be re-used or disclosed to third parties except as required by law, for authorized oversight of the research, or as permitted by patient authorization. All digital and physical patient information will be stored in secure environments (see 2.3.2. Data management). All team members involved in the study will receive proper training

and adhere to confidentiality protocols to protect participant privacy. The study will fully comply with all applicable laws, regulations, and guidelines.

### 3.5. Post-trial care

Ensuring subject safety is a top priority for the research team and hospital staff. In the unlikely chance of an unexpected serious adverse event, the PI, Dr. Pascal Laferrière-Langlois, will be immediately notified and prompt actions will then be taken to provide appropriate care.

### 3.6. Dissemination policy

We aim to publish the trial results in a mid-impact factor journal in the field of anesthesia or medical technology. Patients and the public were not involved in the design, conduct, or reporting of this research, and will not be involved in its dissemination. Only those that contribute significantly to the advancement and publication of the study will be considered eligible for authorship.

## 4. Discussion

As VR emerges as a promising adjunct to intraoperative pharmacological sedation, high-quality research is essential to assess its real-world efficacy, safety, and clinical applicability. While early studies suggest that VR reduces anxiety and sedation requirements during surgery, its widespread adoption in clinical practice must be guided by robust evidence from well-designed trials. This need for rigorous evaluation forms the foundation of evidence-based medicine, ensuring that VR integration into anesthesia protocols is supported by objective data.

The V-RAPS randomized controlled trial will be the first study to evaluate the impact of VR immersion on sedation requirements specifically in a breast surgery population. While previous research has demonstrated positive outcomes in other surgical settings, such as orthopedic surgery, the potential benefits of VR for breast surgery patients remain unexplored. Breast cancer patients, in particular, are highly susceptible to preoperative anxiety and perioperative distress, often describing their experience in terms of "fighting the unknown" and "loss of control" [26]. Given these unique psychological challenges, we believe that exploring VR as an additional anxiolytic modality in this population is of significant clinical interest.

Furthermore, VR technology carries a relatively low risk of adverse effects. The most common anticipated side effect is cybersickness, a form of nausea that can occur with VR immersion. However, previous trials using VR in procedural settings have shown that cybersickness is infrequent and typically mild, with symptoms that resolve quickly following removal of the VR headset and the use of antiemetics if necessary [20,27,28].

While it would be interesting to compare VR immersion to an alternative non-pharmacological distraction method (e.g., music, visual distraction, etc.), we opted for a standard of care control group due to the following rationale. Firstly, a systematic review of intraoperative VR interventions has recently been conducted; in all the trials included in this review, the control group consisted of standard care without any additional distraction techniques [29]. We feel that aligning our methodology with this established approach allows our trial to contribute meaningfully to the existing body of evidence and facilitate its inclusion in future meta-analyses. Furthermore, our primary objective is to evaluate whether VR immersion is beneficial as an adjunct to standard intraoperative care, rather than comparing it to another distraction method, which would answer a different research question. While this is also an interesting question, our current study aims to first establish the potential benefit of VR compared to standard practice in this specific surgical context. As such, the design of our study is pragmatic in nature, focusing on real-world implementation of VR technology in operative settings, including non-randomized scenario selection. Future trials may build upon our findings to compare VR to other non-pharmacological interventions or to formally evaluate differences between VR scenario types.

## 4.1. Limitations of study design

**4.1.1. Unblinded personnel.** Due to the nature of VR interventions, it is difficult for the patient, the anesthesiologist or the surgeon to be blinded to group allocation. We recognize that a sham intervention, such as having a patient wear a non-functional VR headset, could theoretically be considered to achieve blinding. However, we believe this approach presents several important limitations.

Firstly, a non-functional VR headset could potentially induce additional anxiety or feelings of claustrophobia, as it would restrict the participant's vision without providing the immersive, open 3D environment intended to reduce anxiety; This unintended discomfort could, in turn, bias the results of our trial by exaggerating the perceived benefit of the active VR intervention. Secondly, while a sham headset might initially mask group allocation from personnel, it would not effectively blind participants, who would likely realize they are not receiving an active VR experience. This creates a significant risk that participants could inadvertently reveal their allocation during the procedure, which would hamper the effects of the blinding. Lastly, our trial is designed to compare a VR intervention to the current standard of care; a sham headset does not represent standard practice and would therefore lack clinical relevance in this context.

While our trial will not utilize blinding, the use of PCS minimizes the anesthesiologist's influence on propofol administration, reducing performance bias.

**4.1.2. Single-center study.** While the anticipated sample size for our trial is larger than that of many similar studies, which should enhance the robustness and reliability of our findings, this trial will be conducted at a single hospital in Montreal. This design may limit the generalizability of our results to other institutions, patient populations, or healthcare settings.

Therefore, regardless of the outcomes of this trial, future studies with larger sample sizes should be conducted across multiple centers, including different geographical regions and diverse medical practices, to comprehensively evaluate the effectiveness and applicability of VR immersion in intraoperative care. This need for further research is particularly important given that VR-assisted anxiolysis in the operating room remains a relatively novel field, with a body of evidence that is still in its early stages of development.

**4.1.3. Non-randomized VR scenario choices.** One of this study's secondary objectives is the analysis of subgroup outcomes based on the choice of initial VR scenario; specifically, it aims to assess whether interactive or non-interactive scenarios may have a different effect on sedation requirements. However, as previously mentioned, scenario choices will not be randomized, but rather chosen based on participant preference, as this is a pragmatic trial with the goal of eventual real-world implementation. While this grants participants an opportunity to partake in a personalized experience, it may create a form of selection bias in our subgroup analyses. Therefore, we acknowledge that any results arising from such analyses should simply be considered as hypothesis-generating, rather than robust evidence, and will be framed as such.

**4.1.4. Short-term outcomes.** The study focuses solely on short-term outcomes, concluding once participants are discharged from the PACU. While long-term outcomes like post-operative delirium or cognitive impairment were not considered essential for this research, future studies could explore these longer-term effects.

## 4.2. Implementation considerations

Although dedicated research staff will be responsible for managing all aspects of the VR intervention during our trial, we recognize that successful integration of this technology into routine clinical practice will ultimately require training regular hospital personnel in the application and operation of the device. This could involve orderlies, nurses, or even attending anesthesiologists, ensuring that VR-assisted anxiolysis can be seamlessly incorporated into standard intraoperative workflows.

## Supporting information

**S1 Appendix. VRAPS_Appendices.**
(DOCX)

**S1 File. VRAPS_Original_Protocol(Ethics_Approved).**
(DOCX)

**S2 File. VRAPS_SPIRIT_Checklist.**
(PDF)

## Acknowledgments

We thank Paperplane Therapeutics for providing the VR hardware and software. We also thank the whole research team at the Laboratory of Innovative Anesthesia in Montreal (LIAM) for their assistance in research organization.

## Author contributions

**Conceptualization:** Joe Zako, Nicolas Daccache, Julien Burey, Louis Morisson, Pascal Laferrière-Langlois.

**Investigation:** Joe Zako, Julien Burey, Pascal Laferrière-Langlois.

**Methodology:** Joe Zako, Nicolas Daccache, Julien Burey, Ariane Clairoux, Louis Morisson, Pascal Laferrière-Langlois.

**Supervision:** Pascal Laferrière-Langlois.

**Writing – original draft:** Joe Zako, Julien Burey.

**Writing – review & editing:** Nicolas Daccache, Louis Morisson, Pascal Laferrière-Langlois.

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
