## [Decision Letter · Decision Letter 0]

Dear Dr. Zako,

Thank you for submitting your manuscript to PLOS ONE. After careful consideration, we feel that it has merit but does not fully meet PLOS ONE’s publication criteria as it currently stands. Therefore, we invite you to submit a revised version of the manuscript that addresses the points raised during the review process.

We look forward to receiving your revised manuscript.

Kind regards,

Marta Tremolada, Ph.D.

Academic Editor

PLOS ONE

Journal Requirements:

Dr PLL declares ownership interest in private companies unrelated to this work (Divocco Medical and Divocco AI). Other authors declare no competing interests.

Additional Editor Comments:

I suggest to follow the revisions suggested by the three reviewers. The paper is interesting and the topic is quite new, but major revisions should be done.

Reviewers' comments:

Reviewer's Responses to Questions

**Comments to the Author**

1. Does the manuscript provide a valid rationale for the proposed study, with clearly identified and justified research questions?

Reviewer #1: Yes

Reviewer #2: Yes

Reviewer #3: Yes

2. Is the protocol technically sound and planned in a manner that will lead to a meaningful outcome and allow testing the stated hypotheses?

Reviewer #1: Yes

Reviewer #2: Partly

Reviewer #3: Yes

3. Is the methodology feasible and described in sufficient detail to allow the work to be replicable?

Reviewer #1: Yes

Reviewer #2: No

Reviewer #3: Yes

4. Have the authors described where all data underlying the findings will be made available when the study is complete?

Reviewer #1: Yes

Reviewer #2: Yes

Reviewer #3: Yes

5. Is the manuscript presented in an intelligible fashion and written in standard English?

Reviewer #1: Yes

Reviewer #2: Yes

Reviewer #3: Yes

You may also provide optional suggestions and comments to authors that they might find helpful in planning their study.

Reviewer #1: This manuscript is essentially a study protocol to conduct a single-center, randomized controlled trial (RCT) for comparing the impact of Virtual reality (VR) immersion vs controls on IV sedation in adults undergoing breast surgery. The study was registered in clinicaltrials.gov, with a valid NCT number, and approved by the respective Ethics/IRB board. While the objectives and timeliness of this project appear sound and convincing, some comments appear below:

(a) The sample size/power stab doesn't mention the kind of statistical test used, whether the alternative hypothesis was one-or two-sided, and the desired effect size under consideration.

(b) Randomization: Details on the randomization is missing. Will it be a 1:1 randomization? What techniques will be used? Like, block randomization? If block, then what's the block size?

(c) Randomization: How are the blinding and allocation concealment (two separate steps) to be conducted? Again, details missing. To my surprize, this was mentioned as a limitation much later in the manuscript, whereas, the details regarding these should have appeared earlier.

(d) Statistical Analysis: Please replace "multivariate" logistic regression (LR) with "multivariable" logistic regression.

Also, there is no mention of associated goodness-of-fit (GOF) assessments after conducting LR fits, say via Hosmer-Lemeshow GOFs.

(e) Conclusions/Discussion: A well thought out Conclusions/Discussion section is missing. In there, it should be mentioned that the expected results from this study is only for this sample of patients recruited at a Montreal hospital, and future trials with larger sample sizes and at other geographical locations will be necessary to assess the effectiveness of the VR.

Reviewer #2: Thanks for the opportunity to review your manuscript/protocol. See below my major and minor comments

Major comments

1. What is the reason for not first performing a pilot/feasibility study instead immediately proceding to a randomized study. This rationale should be explicitly mentioned and in I would strongly urge you to reconsider, as a pilot study will provide pivotal information essential to more properly perform your randomized study.

Moreover, please provide evidence (or pilot data) that a VR intervention in this population is safe, and does not pose a higher risk for side effects like cybersickness. Finally, add data on the level of immersiveness of your specific VR intervention

2. The control group

As the intervention is aimed as a distraction method, it is likely more informative to include a different (less costly) distraction strategy as comparator. Please clarify the selection of your control group.

3. The intervention group

Is the intervention VR-application fully immersive and includes a headphone. See for example https://pubmed.ncbi.nlm.nih.gov/34079940/ ; please explicitely describe whether this an commercial product or not

The possibility of different VR scenarios is nice, but add a level of variability that will hamper the effect of your intervention ans may result in treatment variation bias: please clarify (reconsider to perform a pilot study to select your most promising scenario)

It is unclear who accompanied patients during the VR intervention, as this maybe particularly relevant to the SoC and intervention effect on the primary endpoint (please take future implementation barriers into consideration)

4. Outcomes

Please provide clarity on how you operationalization your secondary and exploratory endpoints as these are currently partly missing in the manuscript (lines 159-178)

The background of the Nociception Level (NOL) index is poorly mentioned, and as such the rationale to include NOL as exploratory outcome is unclear. Please elaborate

Not including longer-term (patient-centered) outcomes is a missed opportunity (see also L330-331). Please explain why you did not include such relevant outcome (and reconsider)

5. Sample Size

Please elaborate and provide references that substantiate that a 30% reduction in clinically relevant. In addition, provide historical data on propofol requirements during the surgical procedure under study.

6. Consider including the recently published ‘Reporting Guidelines for the Early-Phase Clinical Evaluation of Applications Using Extended Reality: RATE-XR Qualitative Study Guideline’ (PMID: 39612482) and add its items that are currently missing into your manuscript.

Minor comments

L34: 'explore'change into 'determine'

L36 include description of the application used in your abstract

L78-80: add reference to substantiate your claim; moreover, add a sentence on standard anesthesia during this surgical procedure

L85: appreciate change into determine

L104: omit minimal risk (see also L267). Add single-center 1. minimal risk is not part of study design; 2. this is your assumption: please elaborate your arguments that this is a low-risk study

L115: add age cut-off for your adult patients

L120 Eligibility criteria: Severe mental impairment: How operationalized?

L153-154: please provide your pre-specified criteria for switching to general anesthesia

L159-160: Validated method?

L161: please add the rationale to collect data on pre-operative anxiety, and how you measure this (and its cut-off)

L187 please specify 'variety'

L217-218 please clarify the questionnaire used (reference, validated? Self-composed?)

L254-255: please clarify and specify: ‘if relevant‘, ‘additional variables’ what is the outcome of interest?

L255-257: please clarify, and omit secondary. Which subgroups?

L257-258 please clarify what you intend to do?

L263: please provide the level of statistical significance

L275-276: please specific how your screen for, define your side effects (see also L162-164)

L276-277: omit, as already mentioned in the exclusion criteria

Reviewer #3: Thank you for the opportunity to review this protocol manuscript, “Virtual reality as a strategy for intra-operatory anxiolysis and pharmacological sparing in patients undergoing breast surgeries: the V-RAPS randomized controlled trial.” The study is well designed and the protocol follows the SPIRIT guidelines. I have a few points for consideration by the authors.

L 103: Was this definition of minimal risk one used by your IRB or is this your opinion? Often randomized trials that involve drugs, especially self administered sedatives, are considered “full board” and not minimal risk.

In the action scenario, the patients are allowed to move their hands and arms freely?

L163: Define hypotension and hypoxia

L175: Explain why you would not assess satisfaction in the control group. Wouldn’t patient satisfaction with their anesthesia experience be worth comparing?

L257: If the patient requires general anesthesia, is this recorded separately as a secondary outcome? Is the primary outcome data still analyzed?

L261: Provide more details on how you will make a predictive model for NOL spikes and subsequent propofol bolus

L271: Any potential harms associated with patient self administered propofol?

L321: It is possible, however difficult, to blind anesthesia and surgery personnel to the study assignment by using sham equipment and limiting verbal communication about the experience. If you chose not to use blinding in your study, I would recommend changing the phrasing from “it is not possible” to “we decided” etc. An additional limitation is that your study does not control for the placebo effect, which also could have been accounted for by the use of sham equipment.

L338: VR headsets and software cost money. Would you not consider Paper Plane’s provision of these materials to be industry support and worth mentioning in the funding section?

**Do you want your identity to be public for this peer review?** For information about this choice, including consent withdrawal, please see our Privacy Policy

Reviewer #1: No

Reviewer #2: **Yes: ** Evert-Jan Wils

Reviewer #3: No

---

## [Author Response · Author response to Decision Letter 1]

26 Mar 2025

Reviewer #1: This manuscript is essentially a study protocol to conduct a single-center, randomized controlled trial (RCT) for comparing the impact of Virtual reality (VR) immersion vs controls on IV sedation in adults undergoing breast surgery. The study was registered in clinicaltrials.gov, with a valid NCT number, and approved by the respective Ethics/IRB board. While the objectives and timeliness of this project appear sound and convincing, some comments appear below:

(a) The sample size/power stab doesn't mention the kind of statistical test used, whether the alternative hypothesis was one-or two-sided, and the desired effect size under consideration.

Thank you for your comment. The desired effect size was briefly mentioned on page 8:

“For our sample size calculation, we estimated a 30% reduction of propofol usage in the VR group compared to the control group.”

We have now provided more information regarding our planned statistical analysis:

Added section :

“A two-sample t-test was used for the calculation, with a two-sided hypothesis test and a significance level of 0.05. To achieve 80% power, 90 participants were required, with the sample size inflated to 100 to account for attrition (e.g., dropout, protocol deviations).”

(b) Randomization: Details on the randomization is missing. Will it be a 1:1 randomization? What techniques will be used? Like, block randomization? If block, then what's the block size?

Thank you for this comment and we apologize for the oversight. This section has been clarified as such:

“Electronic randomization of the participants will be performed by an independent third party, using the National Cancer Institute’s Clinical Trial Randomization tool. Specifically, we will randomize participants in a 1:1 ratio using a maximal randomization method, allowing a maximum tolerated imbalance of 3.”

(c) Randomization: How are the blinding and allocation concealment (two separate steps) to be conducted? Again, details missing. To my surprise, this was mentioned as a limitation much later in the manuscript, whereas, the details regarding these should have appeared earlier.

Thank you for your comment. Our manuscript already notes that blinding of participants and personnel is not possible due to the nature of the virtual reality (VR) intervention. We recognize that a sham intervention, such as having a patient wear a non-functional VR headset, could theoretically be considered to achieve blinding.

However, we believe this approach presents several important limitations:

A non-functional VR headset could potentially induce additional anxiety or feelings of claustrophobia, as it would restrict the participant’s vision without providing the immersive, open 3D environment intended to reduce anxiety.

While a sham headset might initially mask group allocation from personnel, it would not effectively blind participants, who would likely realize they are not receiving an active VR experience. This creates a significant risk that participants could inadvertently reveal their allocation during the procedure.

Importantly, our trial is designed to compare the VR intervention to the current standard of care. A sham headset does not represent standard practice and would therefore lack clinical relevance in this context.

We have added a passage in the discussion section regarding this comment.

As for the concealment method, it has also been mentioned in the assignment of interventions section:

“Each participant’s allocation will be sealed in an opaque envelope and handed to the dedicated research staff not involved in patient care. The envelope will be opened at the patient's entry into the operating ward, after confirming that the surgery will be performed under regional anesthesia. Due to the nature of VR, the intervention cannot be blinded to participants or personnel after allocation.”

(d) Statistical Analysis: Please replace "multivariate" logistic regression (LR) with "multivariable" logistic regression.

Also, there is no mention of associated goodness-of-fit (GOF) assessments after conducting LR fits, say via Hosmer-Lemeshow GOFs.

Thank you for your comment.

“Multivariate” has been replaced by multivariable.

As for the second comment, we have also added a clarifying passage in the manuscript:

“To evaluate the fit of our logistic regression models, we will conduct the Hosmer-Lemeshow goodness-of-fit test. A p-value greater than 0.05 will be considered indicative of an adequate model fit. If poor fit is detected, alternative modeling strategies, such as transformations or interaction terms, will be considered.”

(e) Conclusions/Discussion: A well thought out Conclusions/Discussion section is missing. In there, it should be mentioned that the expected results from this study is only for this sample of patients recruited at a Montreal hospital, and future trials with larger sample sizes and at other geographical locations will be necessary to assess the effectiveness of the VR.

Thank you for your comment.

The discussion section has been expanded to be more thoughtful and complete.

Reviewer #2: Thanks for the opportunity to review your manuscript/protocol. See below my major and minor comments

Major comments

1. What is the reason for not first performing a pilot/feasibility study instead immediately proceding to a randomized study. This rationale should be explicitly mentioned and in I would strongly urge you to reconsider, as a pilot study will provide pivotal information essential to more properly perform your randomized study.

Moreover, please provide evidence (or pilot data) that a VR intervention in this population is safe, and does not pose a higher risk for side effects like cybersickness. Finally, add data on the level of immersiveness of your specific VR intervention

Thank you for your comment.

We did not consider a pilot or feasibility study necessary in this case, as our trial builds upon an established body of literature with several similar trials already conducted using comparable methodologies. Many studies have shown that VR technology is relatively easy to implement in procedural contexts, and have also shown either equivalent or superior patient satisfaction in the VR group (https://pubmed.ncbi.nlm.nih.gov/38413184/) (https://pubmed.ncbi.nlm.nih.gov/28598921/) (https://pubmed.ncbi.nlm.nih.gov/36129891/) (https://pubmed.ncbi.nlm.nih.gov/32092098/) (https://pubmed.ncbi.nlm.nih.gov/38851457/). Also, we have recently performed a systematic review on intraoperative VR interventions and sedation usage reduction, which yielded promising results in this field (https://pubmed.ncbi.nlm.nih.gov/39862969/). While there are not yet any such studies specifically evaluating breast surgery patients under regional anesthesia, we have previously demonstrated that mastectomies performed under paravertebral block without general anesthesia are a safe and effective anesthetic strategy, providing benefits such as higher turnover rates, reduced postoperative nausea, and faster hospital discharge (https://pubmed.ncbi.nlm.nih.gov/34997554/). At our institution, this approach is now standard practice for mastectomy procedures. Additionally, prior to developing this trial, we informally tested the combination of paravertebral blocks with VR immersion in a small number of patients undergoing breast surgery under regional anesthesia, who tolerated the intervention well and reported high satisfaction.

Given these factors, we believe that a pilot study is unnecessary.

Furthermore, patients undergo thorough screening to exclude those with conditions that could be exacerbated by VR usage, such as epilepsy. Cybersickness, a form of nausea induced by VR immersion, is a relatively well-known and anticipated side effect of this technology. However, previous trials have shown that it is infrequent and typically mild, with symptoms that are easily managed (https://pubmed.ncbi.nlm.nih.gov/28598921/,
https://pubmed.ncbi.nlm.nih.gov/32092098/,
https://pubmed.ncbi.nlm.nih.gov/32597390/).

A brief passage on cybersickness has been added to the discussion.

Finally, the level of immersiveness of the VR intervention will be addressed in the answer to your third comment.

2. The control group

As the intervention is aimed as a distraction method, it is likely more informative to include a different (less costly) distraction strategy as comparator. Please clarify the selection of your control group.

Thank you for your comment.

We acknowledge that comparing VR immersion to an alternative, less costly distraction strategy (e.g., music, guided imagery, or visual distraction) could offer valuable insights. However, we chose the current control group (no VR, standard care) based on two key considerations:

We recently conducted a systematic review of intraoperative VR interventions (https://pubmed.ncbi.nlm.nih.gov/39862969/). In all the trials included in this review, the control group consisted of standard care without any additional distraction techniques. Aligning our methodology with this established approach allows our trial to contribute meaningfully to the existing body of evidence and facilitate its inclusion in future meta-analyses.

Our primary objective is to evaluate whether VR immersion is beneficial as an adjunct to standard intraoperative care, rather than comparing it to another distraction method. Introducing a separate distraction intervention as a comparator would answer a different research question, namely, whether VR is superior to other distraction strategies. While this remains an important question, our current study aims to first establish the potential benefit of VR compared to standard practice in this specific surgical context. Future trials may build upon our findings to compare VR to other non-pharmacological interventions.

We hope this clarifies our rationale for the choice of control group and shows that this decision was made with careful consideration of both the existing evidence and our specific research objectives.

A passage relating to our rationale has been added to the discussion section.

3. The intervention group

Is the intervention VR-application fully immersive and includes a headphone. See for example https://pubmed.ncbi.nlm.nih.gov/34079940/ ; please explicitly describe whether this an commercial product or not

The possibility of different VR scenarios is nice, but add a level of variability that will hamper the effect of your intervention ans may result in treatment variation bias: please clarify (reconsider to perform a pilot study to select your most promising scenario)

It is unclear who accompanied patients during the VR intervention, as this maybe particularly relevant to the SoC and intervention effect on the primary endpoint (please take future implementation barriers into consideration)

Thank you for your comment.

We have clarified the device being used and the level of immersiveness in the interventions section.

“The VR headset to be used is the commercially available Paperplane Therapeutics VR system, specifically designed for medical applications. It offers a fully immersive experience incorporating both visual and auditory elements; notably, it is equipped with integrated near-ear speakers positioned along the side branches of the headset. These speakers are designed to direct sound toward the patient’s ears, providing clear audio at a volume that is perceptible to the patient but remains low enough to avoid disturbing the surgical team or others in the operating room. This design also ensures that patients can continue to communicate freely with clinicians throughout the procedure.”

Concerning the possibility of different VR scenarios, we understand the concern regarding the potential for increased variability and treatment variation bias. However, the inclusion of different scenarios is intentional and aligns with one of our secondary objectives, which is to explore whether passive (non-interactive) or interactive VR experiences are more effective in reducing intraoperative medication requirements. Indeed, prior literature suggests that interactive VR content may have superior anxiolytic properties compared to passive experiences, but this has only been established in the context of dental surgery (http://pubmed.ncbi.nlm.nih.gov/37993645/). Therefore, we found it interesting not only to assess the overall efficacy of VR but also to provide preliminary insights into the impact of scenario interactivity on sedation needs.

While this approach introduces some degree of heterogeneity, we aim to document each participant’s scenario selection and account for this variability in our subgroup analyses, which we believe will mitigate the risk of treatment variation bias. Just as not all movies appeal to all audiences, our study explores a real-world application of VR where patients have autonomy in selecting their VR scenario.

Finally, concerning the person who accompanies patients during the VR intervention, we have added a passage in the interventions section explaining who will accompany the patient:

“A dedicated research staff member will be present in the operating room throughout the procedure to manage all aspects of the VR intervention, including assisting with headset application, adjusting the VR scenarios upon patient requests, and addressing any technical issues or patient discomfort related to the VR system.”

We have also added a passage in the discussion relating to future implementation.

“Although dedicated research staff will be responsible for managing all aspects of the VR intervention during our trial, we recognize that successful integration of this technology into routine clinical practice will ultimately require training regular hospital personnel in the application and operation of the device. This could involve orderlies, nurses, or even attending anesthesiologists, ensuring that VR-assisted anxiolysis can be seamlessly incorporated into standard intraoperative workflows.”

4. Outcomes

Please provide clarity on how you operationalization your secondary and exploratory endpoints as these are currently partly missing in the manuscript (lines 159-178)

Thank you for your comment.

We have added information where required and clarified the operationalization for each individual outcome (see 2.1.4. Outcomes in the manuscript).

The background of the Nociception Level (NOL) index is poorly mentioned, and as such the rationale to include NOL as exploratory outcome is unclear. Please elaborate

A background paragraph on the NOL index with a rationale for the exploration of its use has been added to the introduction, prior to the objectives.

[...] Not including longer-term (patient-centered) outcomes is a missed opportunity (see also L330-331). Please explain why you did not include such relevant outcome (and reconsider)

We recognize the importance of long-term, patient-centered outcomes such as postoperative pain, functional recovery, and patient satisfaction beyond the immediate perioperative period. However, the primary objective of this trial is to evaluate the intraoperative effects of VR immersion on sedation requirements and patient anxiety during surgery under regional anesthesia. As such, our focus is on the immediate intraoperative and early postoperative (PACU) period, where the impact of VR on sedation and acute recovery is most directly observable. We felt that this was most relevant to the immediate goals of VR as a non-pharmacological anxiolytic and sedation-sparing tool in the intraoperative setting.

5. Sample Size

Please elaborate and provide references that substantiate that a 30% reduction in clinically relevant. In addition, provide historical data on propofol requirements during the surgical procedure under study.

Thank you for your comment.

Available evidence suggests that dose reductions in this range can lead to meaningful clinical improvements and have been considered a clinically significant benefit to patients based on previous research. For instance, in a study comparing patient-controlled sedation (PCS) to clinician-controlled sedation found a 39.3% reduction in propofol usage in the PCS group, which correlated with fewer episo

---

## [Decision Letter · Decision Letter 1]

Dear Dr. Zako,

Thank you for submitting your manuscript to PLOS ONE. After careful consideration, we feel that it has merit but does not fully meet PLOS ONE’s publication criteria as it currently stands. Therefore, we invite you to submit a revised version of the manuscript that addresses the points raised during the review process.

We look forward to receiving your revised manuscript.

Kind regards,

Marta Tremolada, Ph.D.

Academic Editor

PLOS ONE

Journal Requirements:

**Additional Editor Comments:**

The paper is really ameliorated following the reviewers' suggestions. Only some minor issues should be addressed following the indications of Reviewer 2.

Reviewers' comments:

Reviewer's Responses to Questions

**Comments to the Author**

1. Does the manuscript provide a valid rationale for the proposed study, with clearly identified and justified research questions?

Reviewer #1: Yes

Reviewer #2: Yes

2. Is the protocol technically sound and planned in a manner that will lead to a meaningful outcome and allow testing the stated hypotheses?

Reviewer #1: Yes

Reviewer #2: Yes

3. Is the methodology feasible and described in sufficient detail to allow the work to be replicable?

Reviewer #1: Yes

Reviewer #2: Yes

4. Have the authors described where all data underlying the findings will be made available when the study is complete?

Reviewer #1: Yes

Reviewer #2: Yes

5. Is the manuscript presented in an intelligible fashion and written in standard English?

Reviewer #1: Yes

Reviewer #2: Yes

You may also provide optional suggestions and comments to authors that they might find helpful in planning their study.

Reviewer #1: The authors were able to address my previous comments with a great degree of satisfaction. I have no further comments.

Reviewer #2: Thanks for the opportunity to review your revised manuscript/protocol. Your adjustments and answers have added clarity to the study design and the rationale of your choices made.

I have only a few minor follow-up comments:

1. As your study compares the VR-intervention with standard of care (control rather than other method), your study is pragmatic by nature. It will not be able to differentiate between an effect of the interface or the content (despite the non-randomized choice for scenario by patients). Consider to add pragmatic into the phrasing of the relevant paragraph (L417-428)

2. Although the secondary objective on (non-interactive) or interactive VR is of interest, I would remain hesitant in general to make adjustments in your study design to facilitate your secondary objective that may hamper robustness of your primary objective (previous major comment 3). Moreover, to account for this variability in intended subgroup analysis remains cumbersome, as it is non-randomized. Please add this into your limitation section.

3. L217-218 please clarify the questionnaire used (reference, validated? Self-composed?). The fact that your self-reported preoperative questionnaire has been screened and validated by your institution’s ethical committee, does not imply that the questionnaires are validated scientifically. Please rephrase custom-designed questionnaire into self-composed, and add a sentence into the manuscript that it is not (yet) validated.

**Do you want your identity to be public for this peer review?** For information about this choice, including consent withdrawal, please see our Privacy Policy

Reviewer #1: No

Reviewer #2: **Yes: ** Evert-Jan Wils

---

## [Author Response · Author response to Decision Letter 2]

19 May 2025

Reviewer #1: The authors were able to address my previous comments with a great degree of satisfaction. I have no further comments.

We sincerely thank you for the time and energy you have invested into the betterment of our manuscript.

Reviewer #2: Thanks for the opportunity to review your revised manuscript/protocol. Your adjustments and answers have added clarity to the study design and the rationale of your choices made.

Thank you very much for your comments and dedication to improving our manuscript; it is very much appreciated.

I have only a few minor follow-up comments:

1. As your study compares the VR-intervention with standard of care (control rather than other method), your study is pragmatic by nature. It will not be able to differentiate between an effect of the interface or the content (despite the non-randomized choice for scenario by patients). Consider to add pragmatic into the phrasing of the relevant paragraph (L417-428)

We thank you for this valuable comment and agree with your assessment. This phrasing has been added accordingly in the relevant paragraph:

“Furthermore, our primary objective is to evaluate whether VR immersion is beneficial as an adjunct to standard intraoperative care, rather than comparing it to another distraction method, which would answer a different research question. While this is also an interesting question, our current study aims to first establish the potential benefit of VR compared to standard practice in this specific surgical context. As such, the design of our study is pragmatic in nature, focusing on real-world implementation of VR technology in operative settings, including non-randomized scenario selection. Future trials may build upon our findings to compare VR to other non-pharmacological interventions or to formally evaluate differences between VR scenario types.”

2. Although the secondary objective on (non-interactive) or interactive VR is of interest, I would remain hesitant in general to make adjustments in your study design to facilitate your secondary objective that may hamper robustness of your primary objective (previous major comment 3). Moreover, to account for this variability in intended subgroup analysis remains cumbersome, as it is non-randomized. Please add this into your limitation section.

Thank you once more for your attention to our secondary objective. You are correct in stating that the results of these subgroup analyses should be interpreted carefully. To reflect this, we have added an additional limitation in our discussion:

“One of this study’s secondary objectives is the analysis of subgroup outcomes based on the choice of initial VR scenario; specifically, it aims to assess whether interactive or non-interactive scenarios may have a different effect on sedation requirements. However, as previously mentioned, scenario choices will not be randomized, but rather chosen based on participant preference, as this is a pragmatic trial with the goal of eventual real-world implementation. While this grants participants an opportunity to partake in a personalized experience, it may create a form of selection bias in our subgroup analyses. Therefore, we acknowledge that any results arising from such analyses should simply be considered as hypothesis-generating rather than robust evidence, and will be framed as such.”

3. L217-218 please clarify the questionnaire used (reference, validated? Self-composed?). The fact that your self-reported preoperative questionnaire has been screened and validated by your institution’s ethical committee, does not imply that the questionnaires are validated scientifically. Please rephrase custom-designed questionnaire into self-composed, and add a sentence into the manuscript that it is not (yet) validated.

Thank you for your comment. We agree that including such information will better contribute to our transparency. For the sake of conciseness, we have chosen to include this information in the “Data collection methods” section of the methodology rather than on L217-218 (in the objectives section), as this would require us to repeat the information for every item that is assessed on a questionnaire, making that section lengthier than it should be.

“We will collect data at baseline, intraoperatively and postoperatively to assess primary and secondary outcomes. Participant socio-demographic information, medical history and current medication, will be collected from their medical chart. Their education level and familiarity with VR technology will also be recorded through a preoperative questionnaire. Pre-operative anxiety levels will be assessed using the APAIS, a six-item questionnaire designed to measure both anxiety and the desire for information before surgery. The scale provides an overall anxiety score ranging from 4 to 20 and an information-seeking score ranging from 2 to 10, with higher scores indicating greater preoperative anxiety and a stronger desire for information, respectively. A postoperative questionnaire will be used to collect data on patient satisfaction. It is of note that, aside from the APAIS, both our preoperative and postoperative questionnaires are self-composed and have not yet undergone formal scientific validation.”

Furthermore, we have added a reference for the first time the Amsterdam Preoperative Anxiety and Information Score (APAIS) was mentioned, in the objectives section.

---

## [Editor Report · Decision Letter 2]

Virtual reality as a strategy for intra-operatory anxiolysis and pharmacological sparing in patients undergoing breast surgeries: the V-RAPS randomized controlled trial protocol

PONE-D-24-46964R2

Dear Dr. Zako,

We’re pleased to inform you that your manuscript has been judged scientifically suitable for publication and will be formally accepted for publication once it meets all outstanding technical requirements.

Kind regards,

Marta Tremolada, Ph.D.

Academic Editor

PLOS ONE
---

## [Editor Report · Acceptance letter]

PONE-D-24-46964R2

PLOS ONE

Dear Dr. Zako,

I'm pleased to inform you that your manuscript has been deemed suitable for publication in PLOS ONE. Congratulations! Your manuscript is now being handed over to our production team.

Kind regards,

on behalf of

Dr. Marta Tremolada

Academic Editor

PLOS ONE